# Grappling with Issues of Motherhood for Women with Schizophrenia

**DOI:** 10.3390/healthcare11212882

**Published:** 2023-11-02

**Authors:** Mary V. Seeman

**Affiliations:** Department of Psychiatry, University of Toronto, Toronto, ON M5S 1A1, Canada; mary.seeman@utoronto.ca

**Keywords:** antipsychotics, child custody, interventions, mothers, parenting, schizophrenia, stigma, support, women

## Abstract

Despite the fact that most persons with schizophrenia find steady employment difficult to sustain, many women with this diagnosis embrace and fulfill the most difficult task of all—motherhood. The aim of this paper is to specify the challenges of motherhood in this population and review the treatment strategies needed to keep mothers and children safe, protecting health and fostering growth. The review addresses concerns that had been brought to the author’s earlier attention during her clinical involvement with an outpatient clinic for women with psychosis. It is, thus, a non-systematic, narrative review of topic areas subjectively assessed as essential to “good enough” mothering in the context of schizophrenia. Questions explored are the stigma against motherhood in this population, mothers’ painful choices, issues of contraception, abortion, child custody, foster care and kin placement of children, the effects of antipsychotics, specific perinatal delusional syndromes, and, finally, the availability of parental support. This review is intended for clinicians. Recommendations are that care providers work collaboratively with mothers, take note of their strengths as well as their failings, offer a wide array of family services, monitor households closely for safety and for treatment adherence, appreciating the many challenges women with schizophrenia face daily.

## 1. Introduction

Through many decades of close involvement with clinical facilities for schizophrenia, especially those organized to specifically address the needs of women, the author of this review encountered a remarkable paradox. The rate of gainful employment among this population was painfully low, in Europe, reportedly ranging between 8% and 35% [1]. In Denmark, over the 35-year span of adulthood, individuals with schizophrenia earn 14% of the wages earned by their age peers [2]. Barriers to steady employment are reported as being: societal stigma, self-stigma, fear of loss of disability benefits, lack of initiative and confidence to search for jobs, cognitive problems, unpredictable moods and behavior, and poor work records. And yet, paradoxically, 50% of women with schizophrenia perform the most difficult job of all. They become mothers, often having to do this demanding job on their own, without guidance or support [2,3,4,5]. It is, therefore, to be expected that these women face tough challenges fulfilling this crucial personal and societal role and that they will require support, training, and assistance.

In the 10-year experience of the Women’s Clinic for Psychosis (age range: 18–85) a significant percentage of clinic members gave birth, but only a very small minority had continuously reared their children. Some lived with their children in the grandparents’ home where the bulk of childcare was left to the grandmother. Some mothers periodically took full charge. A few had husbands, ex-partners, mothers-in-law, or sisters who worked together with the mothers to nurture the children and keep them in the parental home. Most of the children, however, were placed periodically in foster care; maternal visits were limited, and both mothers and children suffered. Many of the infants born to women in the Clinic were taken into care at birth and adopted, to the birth mother’s unending grief. Some of the women in the Clinic became pregnant again and again, with identical results. These were the experiences that led, 20 years after the Clinic closed, to a literature search on motherhood in schizophrenia. The aim of the paper is to investigate the challenges of motherhood in this population and review the steps needed to help keep women with schizophrenia and their children safe from harm, and with an improved quality of life.

## 2. Materials and Methods

This is a narrative review. The inquiry was conducted by searching PubMed with the terms “motherhood” and “schizophrenia”. This yielded 23 results. I specifically searched the publications of Louise Howard and Simone Vigod on schizophrenia because these are two prominent current researchers in this field. This yielded 41 more references. Knowing that the papers of these investigators would cite much of the previous relevant information, I leaned on their citations for further papers. The papers I examined often pointed me in new relevant directions; a snowball effect. Recency was an important factor in my final choice of studies to cite, as was clinical relevance and research quality. The two latter criteria consist of many subjective elements so that my selection of the pertinent literature is admittedly subjective. I consider subjectivity to be important; otherwise, these days, artificial intelligence would do a much better job than humans in amassing the literature on any topic. I apologize for inadvertent omission of potentially critical papers.

## 3. Results

### 3.1. Complex Decisions

Although the precise rate varies from study to study, approximately 50% of women with schizophrenia become mothers, a similar rate to that of women in the general population [6]. Many women with a schizophrenia diagnosis, however, remain unpartnered or live in sequential, unstable relationships, making motherhood doubly difficult [7]. A meta-analysis of 1404 participants in the United Kingdom with a schizophrenia diagnosis (mean age = 39.9), found that only 15.6% were married [8]. The marriage percentage varies, of course, from region to region, dependent on cultural, religious, and economic factors. The percent married is relatively high in India, for instance, where prevailing beliefs are that marriage can cure psychiatric illness [9]. In addition to being mostly single, many women with schizophrenia, for symptom-related and behavior-related reasons, become alienated from their family of origin [10]. Many are periodically homeless [11]. In some parts of the world, they subsist on governmental disability pensions. In other parts of the world, their livelihood depends on charity. When on government support, there is usually supplementation when they become mothers, but the total does not provide for much beyond bare necessities [12].

Most women with schizophrenia are likely, at some point in the trajectory of their illness, to be offered antipsychotic treatment and, for two thirds of those who accept it and adhere to it, the most evident of their psychotic symptoms (delusions and hallucinations) are well-controlled [13]. The other third, and those not in treatment, struggle with often-debilitating symptoms. Beyond delusions and hallucinations, many women with schizophrenia report negative symptoms (apathy, anhedonia, social alienation), cognitive symptoms (problems with attention, memory, analytic skills) [14], as well as depression, and anxiety [15]. Constant symptoms such as these can be incompatible with what Winnicott referred to as “good enough” infant and child care [16], especially in the context of economic insufficiency and meager social support. These are some of the reasons why, given this diagnosis, family, friends, and medical personnel, as well as the women themselves, are wary about the birth of children to mothers with schizophrenia. There is justifiable concern for the safety and well-being of the children because severely ill mothers have difficulties putting, as they must, their children’s needs before their own. Health care providers and relatives also worry about the effect of the added responsibility and stress of motherhood on the mothers’ mental health [17].

During the pregnancy itself, and especially postpartum, the severity of psychotic symptoms tends to increase [18]. There are also high rates of obstetric difficulties in this population [19,20]. The great fear for mothers, as well as for health providers, is that children will not only inherit a susceptibility to schizophrenia but be rendered doubly and triply susceptible due to difficult gestations, birth complications, inadequate parenting, traumatic experiences, poor schooling, and low levels of income support [21].

Furthermore, there is a widespread fear of the effects on mother and child of antipsychotic medication. Many of the drugs used to treat schizophrenia have strong sedative properties and can over-sedate mothers to a degree where they are frequently unable to respond appropriately to their offspring [22,23]. From a research point of view, human parenting behaviors are so varied and complex that it is difficult to investigate the potential impact of drugs on parent effectiveness [24]. Drug effects are more easily studied in rodents where antipsychotics have been shown to markedly interfere with maternal behaviors such as pup retrieval, pup licking, nest building, and pup nursing [25,26,27]. A constant worry for mothers with severe mental illnesses is that antipsychotic side-effects, such as slow movements, clouded thinking, delayed responses, and emotional blandness, will make it seem to observers that they are unable to appropriately care for their children. For these reasons, they may stop taking their drugs, exposing themselves to repeated relapses.

A major concern in the field is that antipsychotic medications during pregnancy exert adverse effects on the later development of children born to mothers with schizophrenia [28]. This is an important concern, but, in order to know whether children are being adversely affected, mental health workers must regularly monitor the health of these children, and this is not being routinely done. Workers rarely inquire about the children and, while they are well aware of their patients’ psychotic symptoms, they are insufficiently acquainted with the extent or quality of their parenting capabilities. Very few patient charts contain information about the physical or mental health of the patient’s children nor, for that matter, about the size or nature of the patient’s support network [29].

Sight is sometimes lost of the fact that many women with disabilities such as schizophrenia do function successfully as mothers [30]. Network assistance from family, friends, volunteers, co-patients, children’s aid agencies, social agencies, and medical/psychiatric support programs are able to make the parenting journey less stressful than otherwise, providing significant benefit to mothers and children. Looking only at diagnosis, it is impossible to determine whether being a mother, constitutes a health risk for the mother or, perhaps, an advantage. This is especially so when the diagnosis is schizophrenia, where symptoms range from mild to severe, constant to intermittent, where some individuals are fully functional while others are barely so, and where some have close family and other supports that act as safety nets. Whenever social/medical personnel decide, based on diagnosis alone, that motherhood is unwise for a client/patient, this decision falls clearly into the realm of unjust bias. It is a common bias. A schizophrenia study by Thornicroft et al. [31] discovered that, when asked, 38% of study participants reported that they were treated dismissively (or worse) by mental health staff when they tried to discuss starting a family.

### 3.2. Pharmacology and Technology to the Rescue

New tools are now available that can make the choice of motherhood, in some ways, easier for women. Should a woman prefer to wait before conceiving, she has a variety of contraceptive and “morning after” choices that no longer rely solely on the co-operation of male partners [32,33]. One important caveat when women with schizophrenia choose contraception is that medical advice be sought because there are potential interactions between hormone prescriptions and some of the psychiatric drugs they are taking [34].

There are other pregnancy choices (not always easily accessible for women with schizophrenia). A woman can freeze her eggs, postponing parenting to when her circumstances or improvements in her health will make her a more competent mother. If women are worried about carrying disorder genes or about the effects of medications on their fetuses, they can now utilize the help of surrogates. If they do not have a male partner, they can take the route of in vitro fertilization [35].

With respect to induced abortion, 50% of the pregnancies in women with schizophrenia are unplanned and 25% of those unplanned pregnancies are terminated because the mother recognizes her inability to care for a child [36]. These are high rates, attributable to a relative lack of knowledge about contraception [37] and also to a high reported rate of sexual coercion in this population [38,39,40]. Even so, depending on jurisdictional regulations, religious denomination, insurance coverage, distance from metropolitan areas, general knowledge, and social capital, abortions are not accessible for many women with schizophrenia.

These are all personal decisions that women with disabilities must make for themselves [41] but, in the context of schizophrenia, decisional capacity is sometimes deficient and personal choices can, in such cases, be overruled by surrogate decision makers. Determinations of capacity are needed when a woman with schizophrenia is considering the pros and cons of an abortion, or of a tubal ligation or of adopting a baby. When she is judged incompetent to make a decision, a dilemma as to the appropriate surrogate sometimes arises. The woman may, for instance, designate her intimate partner, whereas her parents may claim that they are the ones most concerned with her well-being and most aware of the potential consequences of her decision. While surrogate decision makers are asked to decide as would the patient were she well, they are sometimes tempted to act in their own self-interests instead [42], an issue that can lead to serious ethical/legal quandaries [43].

### 3.3. Sexual Issues in Women with Schizophrenia

Women with schizophrenia report a similar degree of sexual desire as other women of comparable age [44], but their opportunities for interpersonal sex are relatively few. Much is unknown because patients’ sexual practices are rarely inquired about during psychiatric visits with schizophrenia patients [45].

It is important to discuss contraception with all women of childbearing age in the context of schizophrenia, as well as topics such as sexual dysfunction and family planning [45,46,47]. One reason why such discussions are important is for protection of women with schizophrenia who may otherwise place themselves at risk by engaging with short-term partners or, indeed, with strangers [48].

Induced abortion can be psychologically traumatic. Bearing a child and giving the child away for adoption is even more difficult. Currently, in some jurisdictions, it is possible for birth mothers to visit with their biological children—which appears to lessen the pain of separation [49]. Adopted children of mothers with schizophrenia who do not know their birth mothers have been known to search for them and eventually locate them. If the mother has severe psychotic symptoms, the reunion may be very traumatic for the child. The effects of such reunions on both parties, mother and child, have not yet been studied.

### 3.4. Mothering with Schizophrenia

As already mentioned, many women with schizophrenia prove to be effective mothers, despite their many challenges [50]. They are able to garner support, make the necessary provisions should they suffer relapse and hospitalization, and be conscientious about the care of their children while also seeing to their own health needs. Nevertheless, they encounter many problems [51,52]. Many studies conclude that these women struggle with parenting tasks and with fear of the possible effects of their illness on their children [53,54,55]. Motherhood, however, is of prime importance to these women, as it is to most women, and becomes a major aspect of their identity [56]. Loss of child custody is an enduring tragedy in their lives [57].

### 3.5. Child Custody

The first year of a child’s life is the most vulnerable period for a woman with schizophrenia in terms of custody loss. Her symptoms may not have recovered from postpartum exacerbation, and, at the same time, this is the time period during which the child is totally dependent on the mother. Child-protection agencies know that there is a significant relationship between parental perinatal mental health problems and the risk of child maltreatment [58,59]. Under the influence of delusional thinking, mothers with psychosis may physically hurt their child [60] or, as a result of negative symptoms or adverse effects of antipsychotics on cognition, mothers may neglect to feed or provide much needed child care. When this occurs, child-protective services (CPS) step in and temporarily (sometimes permanently) remove the child to a foster home. With adequate support, psychiatric liaising and parental training, however, this may not be necessary. Bilson and Martin [61] have reported that child agencies sometimes go too far, and that, in children born in 2009–2010, one of every nine was a CPS parental-child-abuse suspect. The consensus is that this high level of suspicion and intervention is rarely justifiable.

Involvement with CPS can be counterproductive; in the case of parental serious mental illness, detrimental effects on both children and parents have been shown [61]. Custody loss can be prevented not only by attending to a mother’s mental health but also by addressing the contributions to poor parenting by factors such as lack of education, non-existent social support, domestic violence, substance abuse, and poverty [62]. Parents of children subject to care proceedings have complicated health and social needs that need solutions more humane than removing children from the only home they know. With respect to the well-being of the mother, a review of pertinent studies concluded that parental health issues are exacerbated by child removal [63].

Custody challenges may arise outside of those brought by CPS, for example during divorce proceedings. One survey found that approximately one third of 596 parents with serious mental illnesses had experienced familial disputes over their ability to care for their children [64]. The authors recommended that psychiatric rehabilitation practitioners be aware of threats to continued parenting, and, through therapy, parent training, support, and advocacy, bring family members together to prevent unnecessary custody loss.

### 3.6. Kinship Care

Childcare agencies are increasingly advocating kinship care in preference to foster care when they determine that children require placement [65]. In very many instances, grandparents, usually grandmothers, step in to help. Mental illness in a parent is an important reason why children are brought up in their grandparents’ home [66]. This allows for safe and affectionate care and for an uninterrupted relationship with their mother. The rapport between mother and grandmother can, however, be a difficult one in the context of schizophrenia, and children can suffer from the disharmony. The burden on elderly grandparents of caring for two generations can prove onerous, and can sometimes take a heavy toll on their health [67], resulting in three generations needing care.

### 3.7. Perinatal Syndromes

There are several syndromes related to pregnancy that are sometimes seen in women with schizophrenia and are frequently described, mainly in case reports:Delusional denial of pregnancy

It is possible to remain unaware of being pregnant until late into the pregnancy and sometimes even until delivery. This is not uncommon; it is more prevalent in the general population than giving birth to triplets. Delusional denial of pregnancy is different. In delusional denial, the signs and symptoms associated with pregnancy are acknowledged but attributed to factors other than pregnancy, despite medical evidence of a fetus. The denial sometimes continues even after the child is born, with the mother adamantly stating that the child is not hers [68,69]. Continued delusional denial of pregnancy despite antipsychotic treatment may necessitate involuntary hospitalization, as well as curtailment and potential termination of parental rights. This is necessary due to evidence that mothers with these delusions neglect the child and may inflict injury on the child.

Postpartum psychosis and filicide

Filicide, killing one’s infant, is most often seen as a sequela of postpartum psychosis; although, it usually occurs not during the acute psychosis but sometime later, often in conjunction with suicide [70]. Although delusional depression is the diagnosis usually linked to filicide, filicide is sometimes seen in the context of a schizophrenic disorder [71]. The most predictive symptoms are delusions that directly involve the patient’s children. Other warning signs are unprovoked aggressive behavior, suicidal or homicidal ideation, deteriorating judgment, the presence of command hallucinations, and a rapidly changing mental state [72]. An in-depth study from France of 17 maternal filicide perpetrators found that under half suffered from psychotic disorders. In these, the killing was committed either under the influence of hallucinations or delusions or in the context of threat of separation from the children; the threat being sometimes real and sometimes imagined [73].

Delusional pregnancy

Antipsychotic-induced hyperprolactinemia results in amenorrhea, breast swelling and tenderness, galactorrhea, and weight gain—characteristic signs of pregnancy [74,75]. As a consequence, when a woman prone to delusions is treated with antipsychotics that raise prolactin levels, a delusion of being pregnant (especially if this has been a secret, fervent wish, or, perhaps, a fearful apprehension) is not difficult to intuitively understand. Delusional pregnancy can present as a single delusion, or, more commonly in schizophrenia, in association with other delusions. A recent case report describes denial of pregnancy at one time of life and, in the same schizophrenia patient, delusional pregnancy at a later time period [76]. Many cases of delusional pregnancy are over age 45, suggesting that, as women approach the age when they can no longer have a baby, their wish for one becomes overwhelming [77]. Women with delusional pregnancy need to be treated as any delusional disorder is treated, with psychological insight and an increase in antipsychotic dose (which, however, the patient may not want to take if she believes she is pregnant, and which may also increase her pregnancy-mimicking symptoms).

Postpartum delusional misidentification syndrome

A recent report describes eight cases of delusional misidentification syndrome in the postpartum period, a dangerous situation because a misidentified infant is at risk of harm from an ill mother [78]. Such syndromes are rare, but their early recognition can avert tragedies.

### 3.8. How to Support Women with Schizophrenia Who Are, or Are about to Be, Mothers

Economic support

One avenue of effective intervention for women with schizophrenia who are mothers is to improve their economic circumstances [79]. Increases in household income can improve maternal mental health, the quality of parenting, and household well-being. Linking mothers to sources of aid such as loans, foodbanks, budgeting advice, skill development, therapeutic support, enabling them to meet practical needs (for furniture, food, clothing, rent), introducing employment opportunities, and ensuring welfare rights all lead to positive impacts such as crisis resolution, prevention of children’s entry into care, and a strengthening of the mother’s therapeutic alliance with her mental health team. Axford and Berry [79] acknowledge, however, that the evidence for the long-term effectiveness of income support is extremely thin.

Social Support

Bacon et al., in 2023 [80], synthesized 41 studies of parent and practitioner experiences of support for parents with mental health needs. They noted that difficulties mentioned most often by parents, such as financial issues, tended to be absent in the papers they reviewed. Mothers whose children have been removed by CPS report feeling ashamed, hopeless, and increasingly isolated and increasingly exposed to adversity. Services that focus on child protection often neglect the mental health of parents, which leads the authors to recommend integration of child and parental services. Parents reported that CPS interventions were stressful and intrusive but they did appreciate the provision of family case management and of 24 h crisis services. Parents often felt judged, however, and believed that involvement with services exacerbated their mental health problems. Custody loss was a constant threat hanging over their heads. A further recommendation found in the literature was to include parents in decision making as to what was needed to keep children safe and to base decisions on a strength-based, rather than a deficit-based, perspective of parental capacity (expecting the best while being on constant lookout to prevent the worst) [81,82].

Peer Support

Trained and responsible peer support, increasingly utilized in behavioral health services, is very helpful to women as it is often easier to talk to someone who has had similar experiences than to unburden oneself to a professional care provider [83]. Peers can suggest strategies and resources that aid safe and successful parenting. The main recommendations, as discussed in [84], are as follows: (A) To maintain one’s own mental health, which includes a healthy diet, adequate sleep, routine exercise, attending regular mental health and other medical/dental visits, cancer screenings, vaccination appointments, and adhering to all prescribed regimens. (B) To self-monitor for signs of psychotic relapse. Such signs vary among individuals but often take the form of insomnia, lapsed hygiene, and increasing suspiciousness about the meaning of events or of people’s intentions. Recognition of such signs, the ability to increase antipsychotic doses as needed, and 24 h access to a crisis line are vital components of safety. (C) To develop a plan of action should hospitalization be necessary. The children need to know who to contact in an emergency and the proposed contact persons need to be familiar with the plan. The plans need to be shared with the children’s school and doctors and the surrogate care givers need to have school and medical routines written down, as well as the children’s food preferences and allergies, friend’s contact numbers, and favorite toys and activities. (D) To take advantage of available parenting resources (parent training classes, support groups, parent coaching, home visiting, respite services, online courses). (E) To document personal parenting styles—how emotional problems are addressed, how children’s safety is ensured, how limits are set, how conflicts are resolved, and how children are socialized. (F) To learn how to navigate the legal system with respect to child-abuse-reporting mandates, parental rights, and time limits by which goals set by child-protection agencies must be met. From the side of child protection, workers need to be acquainted with the nature of the disabilities presented by mothers with schizophrenia [85], not only the problem areas, but also the competencies, and the areas of life with which schizophrenia does not interfere.

Specific Interventions

Radley et al. [86] offer a scoping review of 34 interventions designed to support parents who suffer from psychosis. The authors conclude that they have all been imported from parental-assistance programs implemented for parents with other disabilities and that very few have been properly investigated for effectiveness in this specific population. Randomized controlled trials are needed to determine the effect of parent buddy systems, parent discussion groups, formal education about parenting, parenting camps, nurse visitors, respite arrangements, family therapy, online courses, and other forms of parenting support, on outcomes of mothers with psychotic disorders and their children.

## 4. Discussion

The evidence from the literature reviewed here supports the existence of risks inherent in being a child of a mother with schizophrenia. But, it emphasizes the fact that the risks do not hold true for all mothers with schizophrenia; in addition, it provides recommendations on how the risks can be attenuated. In the 1960s and 70s, the decades when psychiatrists believed that parents caused schizophrenia, evidence of parental guilt was observed seemingly everywhere. Weakland and Fry [87] found it in the “incongruent communication” of letters that mothers wrote to their sons. This is an example from psychiatry’s past of how a theoretical conviction can interfere with the ability to see clearly. The authors of the 1962 communication only see contradictions in the letters of mothers to sons with schizophrenia but fail to notice the maternal affection that the letters express. Something similar is occurring today in the relatively rigid way mental health practitioners view the capabilities of mothers with schizophrenia.

Recent in-depth interviews with eight parents with schizophrenia (five mothers and three fathers) show a different side of the story [88]. The investigators’ conclusions from these interviews are that practitioners need to support parents with schizophrenia, moving away from risk-focused approaches to targeting stigma, and promoting family-centered, compassionate help, which is a very difficult task. The data suggest that a risk focus is counterproductive, keeping vulnerable mothers from seeking help and accessing the support they need. For reasons of safety, especially when the symptoms expressed are associated with potentially dangerous outcomes, close monitoring is inescapable, but what one attends to while monitoring must include not only maternal deficiencies, but also mothers’ skills, efforts, and triumphs. The views of patients and family members, including children, are very much needed when decisions are made about children of mothers with mental illness.

Table 1 lists the major challenges to effective motherhood encountered by women, as well as potential interventions

## 5. Conclusions

This paper reviews the challenges involved in fulfilling the world’s hardest task—being a good parent, a task that is made exceedingly difficult when the parent is also dealing with a serious psychiatric illness. The challenges are inherent in the illness—schizophrenia—but also in the associated stigma, poverty, social isolation, and adverse life circumstances, which include the constant threat of child removal. There are difficult decisions to be made for mothers with schizophrenia, and a lack of resources and guidance needed to make these decisions effectively. Assistance is available in many potential forms, but not always accessible, and not guaranteed to prove effective for all help-seekers. Recommendations to care providers are to partner with mothers with schizophrenia, appreciate their strengths as well as their frailties, offer a wide array of family services, monitor closely, and take well-earned pride in successful outcomes.

## Figures and Tables

**Table 1 healthcare-11-02882-t001:** Challenges to effective motherhood in schizophrenia and beneficial interventions.

Challenge	Intervention
1. Poverty [11,79]	1a. Income Supplementation 1b. Food Banks 1c. Budgeting Advice 1d. Skills Development 1e. Loans 1f. Job Opportunities 1g. Meeting Housing & Nutritional Needs
2. Delusions/Hallucinations [12,28,34,40,68,69,70,71,72,78,83]	2a. Home Visits 2b. Peer Support 2c. Antipsychotic Medication 2d. Minimization of Adverse Effects 2e. Ensuring Adherence 2f. For Dangerous Delusions (Denial, Misidentification) or Suicide/Homicide risk, Involve Children’s Aid and Institute Kinship care when Possible and Mother’s Hospitalization, Involuntary when Necessary
3. Cognitive Problems, Depression Negative Symptoms, Sedation [13,14,15,43,84]	3a. Home Help 3b. Parent Buddy Systems 3c. Respite Care, Hobbies 3d. Family Mobilization 3e. Safety Precautions
4. Lack of Child Care Experience [83,84]	4a. Parenting Classes 4b. Discussion Groups 4c. Parenting Camps 4d. Modeling & Home Coaching of Parenting Skills 4e. Ensuring Immunizations 4f. Liaison with Pediatrician 4g. Emphasis on Safety, Stimulation, Routine, 4h. Socialization, Positive Feedback, Discipline 4i. Cooperation with Child Support Agencies
5. Family & Social Alienation [64,65,66,84,86,87]	5a. Family Therapy 5b. Psychoeducation 5c. Emphasis on Familial Support 5d. Friendship Groups 5e. Addressing Perceived Stigma
6. Inability to Self-Manage Illness [50,83,84,85]	6a. Peer Support around Symptom and Side Effect 6b. Management, Diet, Sleep, Exercise, Hygiene 6c. Medication Groups 6d. Recognition of Warning Signs of Relapse 6e. Substance Avoidance 6f. Street Proofing & Contraception
7. Attention to Physical Health [83,84,85]	7a. Routine Visits to Family Physician and Dentist 7b. Healthy Nutrition, Sleep Hygiene, Exercise 7c. Vaccinations 7d. Cancer Screening 7e. 24 Hour Access to Crisis Teams

## Data Availability

Search data are available.

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
