# Peer review of "Grappling with Issues of Motherhood for Women with Schizophrenia"

_healthcare, 2023, doi:10.3390/healthcare11212882_

Round 1

Reviewer 1 Report

Comments and Suggestions for Authors

The main question addressed by the research is: What are the challenges of motherhood faced by women diagnosed with schizophrenia and what are the treatment strategies required to ensure the safety, health, and growth of both the mothers and their children?

Here are some suggestiions to improve the manuscript:

In the Introduction emphasize more on why understanding the challenges of motherhood among women with schizophrenia is essential. This could be in terms of societal implications, the well-being of the children involved, or the mental health implications for the mothers.

Briefly hint at the main challenges or topics that the paper will address in the subsequent sections. This can provide a roadmap for readers and pique their interest.

In the Materials and Methods section, here are some specific improvements the authors might consider:

The criteria for selecting articles are mentioned in general terms such as "most relevant," "best quality," and "clinical importance." Authors should provide a more precise definition or benchmark for each of these criteria to make the review more transparent.

Provide a detailed search strategy, including the specific keywords or phrases used in the PubMed search. This would help in replicability and understanding the scope of the search.

Clarly list the inclusion and exclusion criteria for the articles. For example, did they only consider articles in English? Were there specific study types they were looking for (e.g., randomized controlled trials, observational studies)

Concerning the discussion:

The discussion focuses on recent interviews with eight parents with schizophrenia. While these findings are essential, consider discussing other recent studies or viewpoints to provide a more comprehensive understanding of the current perspective on parents with schizophrenia.

Consider a clearer structure for the discussion. Start with historical perspectives, move to the modern viewpoint, then discuss the implications of these views, and conclude with recommendations.

Discuss any limitations associated with the studies or viewpoints mentioned. For example, the in-depth interviews only involved eight parents – what are the limitations of drawing broader conclusions from such a small sample?

Based on the discussion and evidence presented, provide specific recommendations for practitioners or policymakers about how to better support parents with schizophrenia.

Author Response

1. In the Introduction emphasize more on why understanding the challenges of motherhood among women with schizophrenia is essential. This could be in terms of societal implications, the well-being of the children involved, or the mental health implications for the mothers.

1A.  I have expanded the Introduction. Thank you for the suggestion.

2. Briefly hint at the main challenges or topics that the paper will address in the subsequent sections. This can provide a roadmap for readers and pique their interest.

2A. Thank you. I have done that.

3. In the Materials and Methods section, here are some specific improvements the authors might consider:

The criteria for selecting articles are mentioned in general terms such as "most relevant," "best quality," and "clinical importance." Authors should provide a more precise definition or benchmark for each of these criteria to make the review more transparent.

Provide a detailed search strategy, including the specific keywords or phrases used in the PubMed search. This would help in replicability and understanding the scope of the search.

Clarly list the inclusion and exclusion criteria for the articles. For example, did they only consider articles in English? Were there specific study types they were looking for (e.g., randomized controlled trials, observational studies)

3A. Thank you. I have expanded this section. I have tried to make it a little less routinized and more genuine that what is included in Method sections of many reviews.

4. The discussion focuses on recent interviews with eight parents with schizophrenia. While these findings are essential, consider discussing other recent studies or viewpoints to provide a more comprehensive understanding of the current perspective on parents with schizophrenia.

Consider a clearer structure for the discussion. Start with historical perspectives, move to the modern viewpoint, then discuss the implications of these views, and conclude with recommendations.

Discuss any limitations associated with the studies or viewpoints mentioned. For example, the in-depth interviews only involved eight parents – what are the limitations of drawing broader conclusions from such a small sample?

Based on the discussion and evidence presented, provide specific recommendations for practitioners or policymakers about how to better support parents with schizophrenia.

4A. Mention of the study with the 8 participants is meant to convey the message that patients (and their families) need to be listened to (not just professionals). Since this is a review, the main text has discussed  the information, recommendations and limitations of the available literature. The now included Table makes this clearer. I hope that the revisions address your rightful concerns.

Thank you once again. The revisions are in yellow.

Reviewer 2 Report

Comments and Suggestions for Authors

This is a well-written paper. I would like to ask to revise some parts of this work.

(1) Introduction

This section is very short and does not provide sufficient information regarding the relevance of this review. Thus, this section should be expanded.

(2) Materials and method

This section is also very short and does not provide sufficient information regarding the methodology of this study. Thus, this section should be expanded. Please provide data how many article were analyzed, including and excluding criteria, years of published articles, languages of analyzed articles, keywords, etc. This is a narrative review, however, please add more information of your search strategy and your main question in this work.

(3) Results

Long paragraphs are undesirable (for instance, lines 44-72).

I would like to recommend you to present your key and synthesized findings (summary of review) in the form of a table (e.g., with columns: topic, challenges, implications, recommendations, etc.). This will make your paper very easy to read.

The discussion is too short and imbalanced compared to other parts of the article. Please restructure the narration in the paper.

Author Response

(1) Introduction  This section is very short and does not provide sufficient information regarding the relevance of this review. Thus, this section should be expanded.

Thank you. I have expanded it.

(2) Materials and method. This section is also very short and does not provide sufficient information regarding the methodology of this study. Thus, this section should be expanded. Please provide data how many article were analyzed, including and excluding criteria, years of published articles, languages of analyzed articles, keywords, etc. This is a narrative review, however, please add more information of your search strategy and your main question in this work.

Absolutely. This section has been clarified and expanded.

(3) Results  Long paragraphs are undesirable (for instance, lines 44-72).

3A. I have tried to make shorter paragraphs.

I would like to recommend you to present your key and synthesized findings (summary of review) in the form of a table (e.g., with columns: topic, challenges, implications, recommendations, etc.). This will make your paper very easy to read.

3B. Excellent recommendation. This has been done.

The discussion is too short and imbalanced compared to other parts of the article. Please restructure the narration in the paper.

It is true that the Discussion is short because most of the discussion has already occurred in the Results section. I have tried to lengthen it and the addition of the Table helps a lot. Thank you.

The revisions are in yellow.

Round 2

Reviewer 1 Report

Comments and Suggestions for Authors

I would like to thank the Authors for the changes they have done. I have no further comments.

Author Response

Thank you for your vote of confidence.

Reviewer 2 Report

Comments and Suggestions for Authors

(1) Please number the challeges and interventions in Table 1.

(2) Please add references for the challeges and interventions in Table 1.

Author Response

Thank you very much for these recommendations. The numbering and citations are now in the Table.